# The role of medicines and therapeutics committees structure in supporting optimal antibacterial use in hospitals in Uganda: A mixed method study

Isaac Magulu Kimbowa[1]*, Moses Ocan[1], Jackson Mukonzo[1], Mary Nakafeero[2], Jaran Eriksen[3¤a], Cecilia Stålsby Lundborg[3], Jasper Ogwal-Okeng[4], Celestino Obua[5], Joan Kalyango[6¤b]

1 Department of Pharmacology and Therapeutics, Makerere University College of Health Sciences, Kampala, Uganda, 2 School of Public Health, Makerere University College of Health Sciences, Kampala, Uganda, 3 Department of Global Public Health, Karolinska Institutet, Stockholm, Sweden, 4 Office of the Vice Chancellor, Lira University, Lira, Uganda, 5 Office of the Vice Chancellor, Mbarara University of Science and Technology, Mbarara, Uganda, 6 Department of Pharmacy, Makerere University College of Health Sciences, Kampala, Uganda

☉ These authors contributed equally to this work.
¤a Current address: Unit of Infectious Diseases/Venhälsan, Stockholm South Hospital, Stockholm, Sweden
¤b Current address: Clinical Epidemiology Unit, Makerere University College of Health Sciences, Kampala, Uganda
* jakemagulu@gmail.com

**Data Availability Statement:** We have deposited all the minimum data that supported our work on

## Abstract

Although the roles of Medicines and therapeutic committees (MTCs) have been expanding, there is limited information on the role of their structure in optimal antibacterial use in hospitals, especially in low-and-middle-income countries. Our study explored the structure and role of MTC in supporting antibacterial use in regional referral, general hospitals and tertiary private not-for-profit (PNFP) hospitals in Uganda. We conducted an explanatory sequential mixed-method approach with triangulation to explore the structure and functional role of MTCs from August 2019 to February 2020 in hospitals in Uganda. Quantitative data was collected using an interviewer-administered questionnaire among chairpersons or secretaries of MTCs and was analysed using descriptive statistics. We conducted key informant interviews using an interview guide among long-term serving members of MTCs to collect qualitative data which triangulated the quantitative data. The study revealed that sixteen hospitals had successfully established MTCs with an average duration of the MTCs' existence of 5.6 (+2.7) years. The membership of the MTCs varied between 7 and 14, with a median value of 10, and the majority of members in MTCs were pharmacists (15 out of 16) and clinical specialists (13 out of 16). The most frequent subcommittees of the 16 hospitals MTC were supply chain (n = 14), antimicrobial stewardship (n = 13), and infection control (n = 12). Majority (14 out of 16) of the MTCs supported availability and access of antibacterial use by selecting and evaluating antibacterials agents for their formulary lists using established criteria. Additionally, 15 out 16 MTCs conducted antimicrobial stewardship activities to support optimal antimicrobial use. In our study, MTC membership and subcommittees were critical structural components that aided the selection and evaluation antibacterials on

this paper in the open science framework (OSF), DOI 10.17605/OSF.IO/4E2CF.

**Funding:** The study was funded by Makerere University-Swedish International Development Agency (SIDA) collaboration (Sida PI0010). PI was CO The funders had no role in study design, data collection and analysis, decision to publish, or preparation of the manuscript.

**Competing interests:** The authors have declared that no competing interests exist.

**Abbreviations:** ADR, Adverse Drug Reaction; AMS, antimicrobial stewardship; MTC, medicines and therapeutics committees.

hospital formulary lists and they supported optimal antibacterial use through implementing various antimicrobial stewardship activities. There is a need for the Ministry of Health to conduct more training on operationalising MTCs structures in all hospitals.

## Introduction

For almost eight decades, Medicine and Therapeutics Committees (MTCs) have existed in high-income countries with diverse member compositions and roles to facilitate the rational use of medicines, policy development, and the identification of cost-effective medications in hospital settings [1, 2]. Implementing MTCs interventions in low and middle-income countries (LMICs) has exhibited a gradual and sluggish pace [3]. It was anticipated that LMICs would implement national policies for MTCs after adopting the World Health Organization's (WHO) practical manual for MTCs [3]. Nevertheless, it is common for LMICs to establish MTCs in hospitals based on need, with differing levels of structural organisation, membership composition, and duties due to the limitations in available resources [1, 4, 5]. In hospitals, numerous medicines with comparable or identical therapeutic properties necessitate the implementation of MTC structures to facilitate the efficient and effective selection and optimisation of their administration to patients [6]. Stichele et al. [7] claimed that most LMICs without MTCs in their hospitals observed a threefold increase in antibacterial use and suboptimal antibacterial administration [7]. The absence of an MTC has hindered healthcare providers from having a forum for knowledge sharing and professional development. Consequently, the implementation of interventions to enhance the appropriate use of antibiotics in LMICs was impeded [8].

The WHO practical manual recommends that MTCs establish subcommittees dedicated explicitly to managing supply chains, antimicrobial stewardship, and pharmacovigilance [7]. Additional subcommittees may be established to manage quality control and documentation components. Numerous African studies have assessed the effectiveness of committees in terms of their functional roles in optimising the use of antibacterials [4, 9]. In Nigeria and South Africa, the subcommittees of antimicrobial stewardship committees were found to be the most active, primarily due to concerns surrounding the misuse and abuse of antibacterial agents and the consequent increase in the burden of antibacterial resistance (AMR) within hospital settings [4, 9]. Although MTCs facilitate the implementation of antimicrobial stewardship, several studies have questioned their effectiveness in optimising antibacterial use in terms of generating better patient outcomes, improving patient safety and reducing cost and antibacterial resistance, citing the resource-limited settings in which they are often deployed [9, 10]. The effectiveness of MTCs was hindered by a multitude of factors, including but not limited to overburdened committee personnel, insufficient financial resources leading to the lack of established standards for medicine selection, conflicting personal and professional interests, inadequate communication and performance evaluation, and unpredictable national directives [11].

Under the Ministry of Health's auspices over the past two decades, the Ugandan government has endeavoured to establish MTCs in tertiary public regional referral hospitals and PNFP hospitals [12, 13]. The Ministry of Health has continuously involved leaders of hospitals in establishing or enhancing existing MTCs through training hospital healthcare providers on MTCs and implementing antimicrobial stewardship measures [14]. Despite the various initiatives undertaken by governments and development partners to facilitate the establishment of MTC, there is a lack of information regarding the structure and functional role in optimising

antibacterial use [15]. This study aimed to determine the role of MTCs structure in supporting their optimisation of antibacterial use in tertiary hospitals (regional referral and PNFPs) and general hospitals in Uganda. This study's results will support the development and implementation of MTCs structures to optimise antibacterial use in LMIC hospital settings.

## Material and methods

### Study setting

The study was conducted in public health facilities, including ten regional referrals and three general hospitals) and three tertiary-level PNFPs. We issued invitations to 16 regional referral hospitals, 47 general hospitals, and four tertiary PNFPs to determine who operated MTCs. Regional Referrals Hospitals (RRH) and National Referral Hospitals (NRH) offer free health service delivery. Only ten regional referrals, three general hospitals and three PNFPs with MTCs participated in our study.

Regional referral hospitals provide specialised care and engage in medical research and academic education [16]. The PNFP offers more affordable services than the PFP since it receives government funding [17]. Forty primary PNFP hospitals serve community areas, and four tertiary PNFP hospitals serve urban areas with specialised treatment. Data was gathered between August 2019 and March 2020 before national interventions for COVID-19 were implemented.

### Study design

We conducted an explanatory sequential mixed-method study to ascertain the role of the MTC structure in promoting optimal antibacterial utilisation in regional referrals, general hospitals, and PNFP hospitals from August 2019 to February 2020 in Uganda. The first stage of the study entailed implementing a cross-sectional study utilising an interviewer-administered questionnaire among MTC chairpersons or secretaries to collect quantitative data. During the second stage, we conducted key informant interviews to corroborate, converge, or explain the results of the quantitative study. We employed a deductive sequential mixed method design, prioritising gathering quantitative (QUAN) data and supplementing it with qualitative (qual) data [18]. Prior to formulating an interview guide for the qualitative study, it was essential to undertake the collection of quantitative data (QUAN) as a first stage. The conducted qualitative study provided a thorough understanding of the contextual factors and corroborated and explained quantitative data. It addressed certain aspects that a cross-sectional study may not have accounted for regarding the role of the structure of MTC in supporting optimal antibacterial use.

### Quantitative component

**Sample size and sampling procedure.** Invitations were extended to 16 regional referral hospitals, 47 general hospitals, and four tertiary PNFPs to identify the participation of hospitals with MTCs. Only 21 /67 (31.3%) hospitals reported presence of MTCs. Among the 21 hospitals, only 16 (ten regional referral, accepted and granted the study administrative clearance. We selected 16 MTC chairpersons or their secretaries for their wealth of knowledge and experience in the operations of MTCs.

**Questionnaire development ad validation.** We developed the questionnaire sections from the WHO manual of MTCs and a search of previous literature on MTCs and expert opinions on MTCs [3, 9, 15, 19]. The team of experts from epidemiology, pharmacy, pharmacology, and public health experts agreed upon the questionnaire sections on the structure and functional role and verified their completeness, clarity, and relevance before it was piloted in

three hospitals. During pilot tests, we assessed the wording and interpretation of the questions, and the findings informed changes on a couple of the questionnaire items. The final questionnaire (S1 File) had sections on health facilities, sociodemographics, the structure of MTC, function and interventions implemented.

**Variables.**    The outcome variables of this study were the proportion of MTCs with a structure comprised of certain members and subcommittees. The proportion of MTCs whose structure supported optimising antibacterial use in selected health facilities. Regarding the structure of MTCs, chairpersons or secretaries of MTCs were asked questions about MTC membership, subcommittees, and functional roles performed by the MTC. The response to the question required was "yes", coded as one or "no", coded as zero. The MTC role had a total of 13 statements, which we divided into the following sections: evaluating and selecting antibacterial on the formulary list (three items), managing adverse reactions and medication errors (two items), improving rational antibacterial use (antibacterial stewardship (four items), and developing policies and procedures to optimise antibacterial use (4 items). The response to the questions on roles required was "yes", coded as one or "no", coded as zero. The independent variables included hospital facility type (general, regional referral, and private-not-for-profit), teaching affiliation (teaching and non-teaching hospital) and the number of beds available at each hospital.

**Data collection procedure.**    The quantitative data were collected using an interviewer-administered questionnaire among chairpersons or secretaries of MTCs from August to October 2019 among 16 hospitals. Prior to data collection, training was given to eight pharmacists, four medical officers, two nurses, and two hospital statisticians. In each hospital, a research assistant invited participants and conducted the interviewer-administered questionnaire following a standardised data collection procedure. In stage one, an interviewer-administered questionnaire was administered to 16 chairpersons or secretaries of MTCs in the ten regional referrals, three PNFPs and three general hospitals. Each respondent's interview took between 20 and 30 minutes. The survey was entirely voluntary, with no incentives or collection of specific participant identifiers.

**Data management and analysis.**    We double-checked all questionnaires for accuracy and completeness at the end of each day of data collection. During the data cleaning, we dropped any questionnaires containing significant missing data in their variables. EpiDATA management version 4.2 was used to conduct double data entry and validation. STATA 15.1 was used to analyse the data (StataCorp, Texas, USA). We used frequencies and proportions to summarise categorical data and means and standard deviations for continuous data. We performed descriptive statistics to estimate the proportion of the activities performed in MTCs.

### Qualitative component

**Sample size and sampling procedure.**    We purposefully selected eight long-time-serving MTC members (3 from regional referrals, two PNFPs, and two general hospitals) for in-depth interviews. We selected participants whose hospital service spanned multiple committees. We considered their knowledge, experience, and willingness to participate in the research.

**Interview guide development.**    In stage two, we conducted key informant interviews using an interview guide among the longest–serving members of the MTCs. The interview guide was developed using information from previous studies on MTCs and after analysing data from the first stage of the quantitative component [1, 20–22]. The interview guide comprised open-ended questions centred on explaining the major themes generated from the quantitative component. They also explained the structure of MTC, its functional role in optimising antibacterial use, and other interventions conducted to optimise antibacterial use. The

interview guide (S2 File) was pilot tested among two MTC members in three hospitals to ensure validity and content uniformity.

**Data collection procedure.** We conducted face-to-face key informant interviews using an English guide from October 2019 to February 2020. All interviews were audio-recorded after the interviewee's authorisation. Only the qualitative data that corroborated, explained, or elaborated our first phase of the quantitative study was the focus of our efforts during the interviews. The interviews took an average of 20–30 minutes, and we performed in a private room.

**Qualitative data analysis.** We recorded interviews, transcribed them word-for-word, and checked them for accuracy before qualitative data analysis. The principal investigators (KIM and OM) read the written transcripts (S3 File) several times to familiarise the data and deeply understand it. We coded the transcribed data using an interactive classification and reclassification procedure of codes and subcodes in conjunction with a theme analysis technique. Codes emerged from all relevant words and highlighted sentences, which were later analysed using triangulation. During the discussion with the other authors, we considered codes appropriate for corroboration, explanations or converging information on the role of MTC structure in supporting optimal antibacterial use. The principal investigator met to discuss, compare and refine the coding frame and ensure complete coverage of the data and consistency in the interpretation of the data. We discussed any discrepancies that arose until we reached a consensus. Triangulation ensured that the issue of internal validity was addressed by comparing the findings of the study's quantitative (QUAN) and qualitative (qual) components to determine convergent and divergent findings, as well as additional information offered by one approach in comparison to another.

## Ethical considerations

The Makerere University School of Biomedical Sciences Higher Degree Research and Ethics Committee (SBH-HDREC-624) and the Uganda National Council of Science and Technology (HS339ES) granted their ethical clearance for our study. The study got additional ethical clearance from the Heads of health facilities and granted the study protocol administrative clearance. Before participating in the key informant interview, all targeted respondents gave consent. All information about the key informants was de-identified to ensure anonymity.

## Results

### Characteristics of respondents

Most respondents of MTCs were pharmacists (13/16, 81.3%), having a bachelor's degree as their highest academic qualification and a mean age and standard deviation (SD) of 36.1 (1.1) years. Regional referral hospitals had the highest frequency of MTCs (10/16, 63%) compared to PNFP and General hospitals. Teaching hospitals had the highest frequency of MTCs (12/16, 75%) compared to non-teaching hospitals. Hospitals with a bed capacity of 101 to 350 beds and those with more than 350 beds had the highest frequency of MTC (Table 1).

### Conformity of MTC establishment and their structures to WHO guidelines

**MTC administration.** The average duration of the hospital MTCs' existence was 5.6 (+2.7) years. Fourteen out of the 16 hospitals (88%) established MTCs with a clear goal and purpose, and regional referral centres exhibited the highest level of compliance (90%) to the WHO practical tool guide recommendation. MTCs were incorporated in the hospital's organisational structure (organogram) in 10 out of 16 (63%) of the participating hospitals, where the highest compliance to WHO recommendations was found in all three PNFP hospitals (100%).

**Table 1. Characteristics of the study population (N = 16).**

| Description | Frequency | Percentage |
|---|---|---|
| | (N = 16) | 100% |
| **Type of cadre** | | |
| Pharmacists | 13 | 81.25 |
| Medical specialist | 3 | 18.75 |
| **Type of health facility** | | |
| Regional referral hospitals | 10 | 62.5 |
| General (District) hospitals | 3 | 18.75 |
| Private not for profit (Tertiary) | 3 | 18.75 |
| **Teaching status of the hospital** | | |
| Teaching hospitals | 13 | 81.25 |
| Non-teaching hospitals | 3 | 18.75 |
| **Region** | | |
| Central | 5 | 31.25 |
| East | 4 | 25.0 |
| North | 4 | 25.0 |
| West | 3 | 18.75 |
| **hospital bed capacity** | | |
| 100 beds | 3 | 18.75 |
| 101 to 350 | 7 | 43.75 |
| 351 and above | 6 | 37.5 |

Of the 16 hospital MTCs, 13 (81%) appointed members to the committee and established terms of reference. The highest level of conformity in appointing members was observed in 9 out of 10 regional referrals (90%) and three out of 3 (100%) PNFPs. More than half (56%) of the MTC held meetings once every two month (Table 2).

*"We follow the WHO MTC guidelines when selecting the MTC membership".......KI02. 'The administration appointed new members and gave them a three-year contract indicated on their terms of reference"......KI03. "The hospital already acknowledges the MTC as an important committee and has been included in the hospital administration structure".... KI03. We always share an MTC report with the director and administration on our progress and what kind of support we need"......KI02.*

**MTC membership representation.** The hospitals chose members of various experts (clinical specialists, medical officers, nursing representatives, a microbiologist, finance and administration, as representatives of MTC) with the highest conformity to WHO practical tool standards by regional referral hospitals. The number of members in each MTC ranged from seven to fourteen, with ten members being the median. Pharmacists were the predominant represented group in nearly all MTCs at the participating institutions (n = 15, or 93%), followed by clinical specialists (n = 13, or 81%). Most MTC Chairpersons (n = 12, 75%) were clinical specialists, whereas most (n = 14,88%) MTC Secretaries were pharmacists. (Table 2).

*"We welcome every staff member as long as they are committed and interested in performing the MTC's role. Senior clinicians are no longer the main target for membership; junior medical officers, pharmacists, and nurses are eligible for membership"......KI01. "Members of the*

**Table 2. Structure and composition of MTCs in selected hospitals (N = 16).**

| Structure | Regional referral | PNFP Hospital | General Hospital | Total |
|---|---|---|---|---|
| | n = 10 | n = 3 | n = 3 | N(%) 16 (100%) |
| **MTC administration** | | | | |
| **1.** The hospital administration appointed members to the MTCs and gave them terms of reference. | 9 | 3 | 1 | 13(81) |
| **2.** Does the committee have an established place in the hospital's administration structure | 7 | 3 | 0 | 10(63) |
| 3. MTC activities and interventions supported by hospital administration or management? | 4 | 2 | 0 | 6(38) |
| 4. MTC is established with defined goals and purpose in this hospital | 9 | 3 | 2 | 14 (88) |
| 5. Meetings of MTC | | | | |
| a) Once every month | 2 | 1 | 1 | 4(25) |
| b) Once every two month | 6 | 2 | 1 | 9(56) |
| c) Once every three month | 2 | 0 | 1 | 3(19) |
| **Membership representation on the MTC committees** | | | | |
| **1) Position of the chairperson in the MTC** | | | | |
| a) Clinical Specialist | 8 | 2 | 2 | 12(75) |
| b) Pharmacist | 2 | 1 | 1 | 4(25) |
| **2) Position of the secretary in the MTC** | | | | |
| a) Clinical Specialist | 1 | 0 | 1 | 2 (13) |
| b) Pharmacists | 9 | 3 | 2 | 14 (88) |
| **3) Membership in the MTC** | | | | |
| a) Clinical Specialists (Physicians, Pediatrician, Surgeons, Physicians and Obstetrics and gynaecologists) | 10 | 3 | 0 | 13(81) |
| b) Medical officers | 8 | 1 | 3 | 12(75) |
| c) Pharmacologist | 1 | 0 | 0 | 1(6) |
| d) Pharmacist | 10 | 3 | 2 | 15(94) |
| e) Nursing representatives | 9 | 3 | 1 | 13(81) |
| f) Laboratory specialist | 8 | 1 | 0 | 9(56) |
| g) Finance and Administration | 8 | 2 | 0 | 10(63) |
| **Number of subcommittees** | | | | |
| 0 to 2 | 0 | 0 | 1 | 1 (6) |
| 3 to 4 | 8 | 2 | 2 | 12(75) |
| 5 to 6 | 2 | 1 | 0 | 3(19) |
| **5. Subcommittees of MTC** | | | | |
| a. Supply chain management and logistics committee | 10 | 3 | 1 | 14(88) |
| b. Antimicrobial stewardship committee | 9 | 3 | 1 | 13(81) |
| c. Infection control committee | 9 | 3 | 0 | 12(75) |
| d. Pharmacovigilance committee | 7 | 1 | 1 | 9(56) |
| e. Other | 2 | 2 | 1 | 5(31) |

*MTCs come from all of the major departments of the hospital, including medicine, paediatrics, surgery, obstetrics, and pharmacy. The head nurse has to be there, the laboratory head and then someone in administration". . . . . . . .KI02.*

**MTC subcommittees.** Almost every MTC had at least one subcommittee, with most participating hospitals (n = 12, or 75%) having between three and four subcommittees. The most commonly reported subcommittees were supply chain management committees (n = 14,

88%), antimicrobial stewardship committees (n = 12, 75%), infection control committees (n = 11, 69%), and other subcommittees (n = 5, 3%), such as quality assurance, blood and blood products. Regional referral and PNFP hospitals had a higher compliance rate (above 90%) with the WHO practical tool recommendations in establishing subcommittees on supply chain management, antimicrobial stewardship, and pharmacovigilance than general hospitals (Table 2).

*"To ensure the MTC's effectiveness, we established subcommittees to carry out the mandates for the MTC".……. KI01. We have a subcommittee, the Antimicrobial Stewardship Committee, also called AMS, whose goal is to promote the appropriate use of antibiotics and minimise the development of antibacterial resistance.………KI02. Our supply chain or logistics committee is responsible for managing the supply of medicines on our formulary list".……..KI03. "We usually discuss supply chain issues concerning planning, ordering, distribution and use of medicines"..……..KI01*

## MTC's functional role in optimising antibacterial use in the selected hospitals

**Evaluation and selection of antibacterials for formulary list.** Most MTCs (n = 14,88%) evaluated antibacterials using defined criteria. (Table 3).

*"We focus on several goals while selecting antibacterials, one of which is to reduce maternal and childhood mortality. "We analyse drug consumption data twice a year using ABC analysis. The report will show us how we use antibacterials in the wards.………….KI02. AMS committee adopted the WHO categorisation of antibiotics using AWaRe criteria into hospitals into three groups. "All antibacterials are evaluated for selection using AWaRe criteria "………KI02. We are trying to minimise stockouts in mothers and children of essential medicines in the hospitals. We also ensure we have lifesaving medicines for all categories of patients at all times.………KI01.*

**Implementing antimicrobial stewardship activities to improve the rational use of antibacterials.** Several hospital MTCs implemented AMS activities to optimise antibacterial use. The most (n = 15, 94%) implemented AMS activity was preauthorisation and approval, where a designated clinician was selected to approve the use of antibacterials.

*"One is the major function of the MTC rational use of medicines to improve clinical outcomes and reduce hospital wastage, ensuring appropriate use of antibiotics to minimise antimicrobial resistance development.……..KI01.*

They implemented supplemental AMS strategies to support and optimise antibacterial use, such as; educating healthcare workers on appropriate antibacterial use through CME and small group sessions (n = 15, 94%), conducting periodic evaluations of the hospital's antibacterial use and providing feedback (n = 13, 81%) and updated and implemented standard treatment guidelines and treatment algorithms to followed when managing infectious diseases (n = 14, 88%).

**Table 3. MTC functional role in optimising antibacterial use in children under five selected hospitals (N = 16).**

| | Regional referral hospitals | PNFP hospitals | General hospitals | Total |
|---|---|---|---|---|
| | n = 10 | n = 3 | n = 3 | N = 16 |
| **Evaluating and selecting antibacterials for the formulary list** | | | | |
| MTC has established a procedure for receiving requests for the addition or deletion of antibacterials on the formulary list | 10 | 2 | 1 | 13(81) |
| MTC disseminates information to all members of staff on newly added or deleted medicines on the formulary | 8 | 2 | 1 | 11(69) |
| MTC selects antibacterials based on defined criteria (essential medicine list, standard treatment guidelines, antibacterial resistance patterns) | 9 | 3 | 2 | 14(88) |
| MTC has established a procedure of using antibacterials, not on the formulary | 6 | 2 | 0 | 8(50) |
| MTC has established a procedure for managing drug product shortages, especially antibacterials | 8 | 3 | 1 | 12(75) |
| MTC conducted regular audits on the stock inventory of movement of antibacterials and informed healthcare providers on the use | 7 | 3 | 0 | 10(63) |
| **Improved safety of antibacterials through pharmacovigilance** | | | | |
| MTC has identified, assessed, and reported on adverse drug reactions and medication errors from antibacterials | 7 | 3 | 1 | 11(69) |
| **Implemented antimicrobial stewardship activities to optimise antibacterial use** | 10 | 2 | 2 | 14 (88) |
| MTC educates healthcare professionals through CME and small group meetings on appropriate antibacterial use to improve clinical outcomes and safety. | 10 | 3 | 2 | 15(94) |
| Updated and developed standard treatment guidelines and treatment algorithms to follow when managing infectious diseases | 8 | 2 | 0 | 10 (63) |
| MTC periodically evaluates the use of medicines in the hospital, adherence to standard treatment guidelines and provides feedback on medicine use problems in the hospital | 9 | 2 | 2 | 13(81) |
| MTC conducts developed a policy for the restriction and approval of reserved antibacterial | 10 | 3 | 2 | 15(94) |
| MTC developed structured antibacterials forms or preprinted order forms for use when ordering or dispensing certain antibacterials in specific conditions and on certain wards | 8 | 0 | 0 | 8(50) |
| MTC has developed guidelines for dispensing antibacterial that restrict prescription by qualification | 5 | 1 | 0 | 6(38) |
| **Development of drug policies, standard operating procedures** | | | | |
| Implements a policy to ensure the availability of high-quality antibacterials in the hospitals | 10 | 3 | 1 | 14(88) |
| Policy to assess, track, monitor, track and regulate expenditures on medicines like antibiotics | 10 | 3 | 2 | 15(94) |
| Policy on drug promotion and guidance of Pharmaceutical representatives influence promotion | 8 | 3 | 0 | 11(69) |

*"We have also instituted restrictions on our reserved antibiotics in the second and third line. . . . . ..KI03."* *"We introduced preauthorisation or restriction as an antimicrobial steward-ship strategy where the only medical specialist would access or approve the use of reserved third-line antibacterial agents. . . .. .KI02."* *"We plan to produce antibiograms with the stew-ardship committee to look at all the ward resistance patterns and then disseminate the results quarterly. . . . . . . .KI02".* *"The UCG (Uganda clinical guidelines) is used in specific conditions that need to be treated using particular antibiotics. . . . . ..KI04.'* *"Most members were trained on implementing antimicrobial stewardship intervention to combat antimicrobial resistance, but the workload has been a limitation since these roles of MTC are assigned to us outside our core duties. . . ..KI04*

## Discussion

The results of our study indicate that healthcare facilities at the tertiary level, specifically regional referral and PNFPs hospitals, exhibited a higher number of MTCs compared to

general hospitals. The level of adherence to the WHO practical manual guidelines exhibited heterogeneity across hospitals with MTCs. Unlike general hospitals, tertiary-level facilities (regional referrals and PNFPs) often formally designate members from diverse professional disciplines with contracts and terms of reference. The members varied between 7 and 14, with a median of 10 in each MTC. Tertiary-level healthcare institutions exhibited a higher membership count as compared to general hospitals. Unlike general hospitals, tertiary-level facilities exhibited more well-established subcommittees due to members from diverse professional backgrounds. The subcommittees most commonly encountered in the MTC were supply chain management, antimicrobial stewardship, infection prevention and control, and pharmacovigilance. The role of optimising antibacterial use involved selecting and evaluating antibacterial agents from formulary lists using defined criteria and implementing antimicrobial stewardship programmes.

The credibility, reputation, and legitimacy of MTC among hospital staff can be attributed to the diverse professional backgrounds and extensive experience of its members in managing and accomplishing hospital goals and objectives specific to MTC [9, 23]. The study involved selecting members from various departments, including medicine, surgery, paediatrics, pharmacy, laboratory, and administration by MTC. It was observed that tertiary-level hospitals had more clinical experts and pharmacists than general hospitals. The study found that tertiary hospitals, including regional referrals and PNFPs, exhibited a greater quantity and variety of multidisciplinary team members than general hospitals. The selection of chairpersons for MTCs was based on the clinical professionals who held the most prominent positions while pharmacists served in a secretarial capacity. The results of the structure and membership of our study are consistent with those of prior studies conducted in Nigeria [4]. The results of our study indicate that tertiary-level facilities' MTCs demonstrated greater adherence to the WHO practical manual guidelines regarding membership selection and composition compared to general hospitals. The discrepancy in membership can be attributed to the differences in facility sizes, as tertiary hospitals typically have a bed capacity ranging from 350 to 600, while general hospitals typically have 100 beds. The tertiary-facility level has witnessed a greater frequency of MTC implementation training compared to general hospitals, indicating the possibility of extending such opportunities to the latter as well.

Previous studies have shown that subcommittees are the structures of MTCs that have evolving roles and specialisations depending on the goals of MTCs [24, 25]. Our study agreed with this finding since most of the cardinal roles of MTCs were performed by subcommittees whose membership and competence depended on the size of hospitals. According to our study, supply chain management and logistics, infection prevention and control, antimicrobial stewardship, and pharmacovigilance were the most encountered subcommittees in tertiary-level facilities compared to general hospitals in Uganda. Our findings are comparable to a previous study in Spain which reported a broad representation of subcommittees on medicines selection and infection control [24]. In our study, subcommittees were involved in evaluating and selecting antibacterials, pharmacovigilance, and implementing antimicrobial stewardship activities in all the MTCs. Our findings revealed that MTC in Uganda prioritised the specific subcommittees like supply chain management and antimicrobial stewardship found in all participating facilities compared to pharmacovigilance which was frequent in tertiary-level facilities. The above subcommittees indicated that most Ugandan hospitals aimed to increase access and availability and optimise the use of antibacterials in Uganda. This finding indicates that most hospitals were on a mission to fulfil the goal of the national medicines policy, which emphasises the selection of high-quality medicines, increasing availability and appropriate use [26].

Our findings showed that the core role of MTC was increasing access, availability and appropriate use of maternal and child essential medicines. In our study, MTCs evaluated and selected antibacterials to the formulary using defined criteria, implemented antimicrobial stewardship activities and performed pharmacovigilance. To evaluate and select antibacterials, tertiary-level facilities compared to general hospitals used the AWaRe categorisation alongside the budgetary control and cost-effectiveness analysis. The WHO introduced the AWaRe categorisation of antibacterials as a revision of the model List of Essential Medicines for Children (EMLc) in 2017 [27]. By categorising antibacterials as access, watch, and reserve, the WHO Model List of Essential Medications for Children (EMLc) guided on using Access antibacterials as first- or second-line treatment for severe infections and had to be of high-quality formulations, widely available at a reasonable cost. The categorisation would guide and quantify the demand for antibacterials in healthcare systems and strengthen paediatric antimicrobial stewardship [27]. Our findings on the use of the AWaRe categorisation agreed with a recent study in 2021 in Uganda on national antimicrobial consumption, where the AWaRe categorisation assesses and evaluate antibacterial demand per hospital in-patient admissions [28]. According to Namugambe et al., most of the antibacterials supplied to public health facilities (national referral, regional and general hospitals) were from the access category, followed by the watch category and a small proportion of the reserve antibacterials were stocked in the national and regional referral hospitals. The pharmaceutical budget controls guided MTC to stock a few antibacterials agents in large quantities, thus improving their availability and minimising stockouts. Our findings were similar to studies conducted in Nigeria and Thailand, where their MTCs evaluated and reviewed medicines for their formulary list and participated in implementing antimicrobial stewardship [4, 11]. Our findings concurred with those of Namugambe et al., who found that tertiary hospitals used the AWaRe criteria more frequently than general hospitals, necessitating further training and assistance for the latter. While our study did not specifically address the availability of antibacterials used in the AWaRe criteria, it did indicate that MTC relies on the AWaRe criteria for selecting and evaluating antibiotics.

In our study, MTCs demonstrated that they did not control the selection of antibacterial but also supported the optimal use of antibacterials through implementing different antimicrobial stewardship activities. They supported several antimicrobial stewardship activities to optimise antibacterial use, including educating healthcare providers through CMEs, preauthorisation, and restricting reserved antibacterials. Our findings were confirmed by previous studies in Nigeria, where CME preauthorisation was conducted as an antimicrobial stewardship strategy [29]. MTCs in our study have demonstrated the existence of structures in terms of members, subcommittees and administrative aspects that support the optimising of antibacterial use. There is a need to characterise antimicrobial stewardship programmes in the three types of health facilities.

## Limitations and strengths of this study

Since this study was confined to hospitals with MTCs, our findings cannot accurately represent the other structures in Uganda's hospitals used to optimise antibacterial use. This study employs an explorative mixed-method approach to highlight the structure and functional role of MTCs in optimising antibacterial use in Ugandan hospitals. Our sample sizes were small, but the combination of quantitative and qualitative data ensured that sufficient data was collected to answer our research question. However, it was limited by the number of participating hospitals with MTCs since most general hospitals had not yet established their MTCs. To obtain reliable information, we used a non-probability sampling strategy to select chairpersons, secretaries, long term members of MTCs, and this limited the study in a way that

increased selection bias as well as being responsible for the small sample size for both the qualitative and quantitative components of the study. As a result, conclusions from this study may not apply to other MTCs in low-income contexts. The study participants' responses may differ from those of other MTC members or hospital staff, thus necessitating additional research to understand their perspective on the structure and functional role of the MTCs. However, this study's findings provide insight into existing structures and antimicrobial stewardship practices existent and implemented in public hospitals and PNFPs in Uganda that can support objective four of the national action plan on antimicrobial resistance of establishing antimicrobial stewardship programmes.

## Conclusion

Although the structure and membership of MTCs vary in tertiary-level facilities and general hospitals of Uganda, committees play a critical role in optimising antibacterial use by evaluating and selecting antibacterials for the formulary list and implementing various prominent antimicrobial stewardship programmes in all hospitals. There is a need for further studies to evaluate the attitude and practices on antimicrobial stewardship and the characteristics of antimicrobial stewardship programmes.

## Supporting information

**S1 File. Questionnaire.**
(DOCX)

**S2 File. Interview guide.**
(DOCX)

**S3 File. Transcripts of the interview.**
(DOCX)

## Acknowledgments

The authors acknowledge the hospital administration and the medical staff working in regional referral hospitals, general hospitals and PNFPs for their support and relevant inputs for the study accomplishment. I also thank Dr Godfrey Bbosa and Dr Enos Kitambo for their help and support while analysing qualitative data.

## Author Contributions

**Conceptualization:** Isaac Magulu Kimbowa, Moses Ocan, Jackson Mukonzo, Jaran Eriksen, Cecilia Stålsby Lundborg, Jasper Ogwal-Okeng, Celestino Obua, Joan Kalyango.

**Data curation:** Isaac Magulu Kimbowa, Moses Ocan, Jackson Mukonzo, Mary Nakafeero, Jaran Eriksen, Cecilia Stålsby Lundborg, Jasper Ogwal-Okeng, Celestino Obua, Joan Kalyango.

**Formal analysis:** Isaac Magulu Kimbowa, Moses Ocan, Jackson Mukonzo, Mary Nakafeero, Jaran Eriksen, Cecilia Stålsby Lundborg, Joan Kalyango.

**Funding acquisition:** Cecilia Stålsby Lundborg, Celestino Obua.

**Investigation:** Isaac Magulu Kimbowa, Moses Ocan, Jackson Mukonzo, Jaran Eriksen, Cecilia Stålsby Lundborg, Jasper Ogwal-Okeng, Celestino Obua, Joan Kalyango.

**Methodology:** Isaac Magulu Kimbowa, Moses Ocan, Jackson Mukonzo, Jaran Eriksen, Cecilia Stålsby Lundborg, Jasper Ogwal-Okeng, Celestino Obua, Joan Kalyango.

**Project administration:** Jaran Eriksen, Cecilia Stålsby Lundborg, Celestino Obua, Joan Kalyango.

**Resources:** Jaran Eriksen, Cecilia Stålsby Lundborg, Joan Kalyango.

**Software:** Joan Kalyango.

**Supervision:** Moses Ocan, Jaran Eriksen, Cecilia Stålsby Lundborg, Jasper Ogwal-Okeng, Celestino Obua, Joan Kalyango.

**Validation:** Isaac Magulu Kimbowa, Moses Ocan, Jackson Mukonzo, Mary Nakafeero, Jaran Eriksen, Cecilia Stålsby Lundborg, Jasper Ogwal-Okeng, Celestino Obua, Joan Kalyango.

**Visualization:** Isaac Magulu Kimbowa, Moses Ocan, Jackson Mukonzo, Mary Nakafeero, Jaran Eriksen, Cecilia Stålsby Lundborg, Jasper Ogwal-Okeng, Celestino Obua, Joan Kalyango.

**Writing – original draft:** Isaac Magulu Kimbowa.

**Writing – review & editing:** Moses Ocan, Jackson Mukonzo, Mary Nakafeero, Jaran Eriksen, Cecilia Stålsby Lundborg, Jasper Ogwal-Okeng, Celestino Obua, Joan Kalyango.

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
