## [Decision Letter · Decision Letter 0]

5 Sep 2023

PONE-D-23-21276The role of medicines and therapeutics committees structure in supporting optimal antibacterial use in hospitals in Uganda: A mixed method studyPLOS ONE

Dear Dr. Kimbowa,

Thank you for submitting your manuscript to PLOS ONE. After careful consideration, we feel that it has merit but does not fully meet PLOS ONE’s publication criteria as it currently stands. Therefore, we invite you to submit a revised version of the manuscript that addresses the points raised during the review process.

We look forward to receiving your revised manuscript.

Kind regards,

Joseph Olusesan Fadare

Academic Editor

PLOS ONE

Journal Requirements:

"No author with competing interest"

Reviewers' comments:

Reviewer's Responses to Questions

**Comments to the Author**

1. Is the manuscript technically sound, and do the data support the conclusions?

Reviewer #1: Yes

Reviewer #2: Partly

2. Has the statistical analysis been performed appropriately and rigorously? 

Reviewer #1: Yes

Reviewer #2: N/A

3. Have the authors made all data underlying the findings in their manuscript fully available?

Reviewer #1: Yes

Reviewer #2: Yes

4. Is the manuscript presented in an intelligible fashion and written in standard English?

Reviewer #1: Yes

Reviewer #2: No

5. Review Comments to the Author

Reviewer #1: Thanks for the authors work on this interesting snapshot of the MTCs structure and function in their country. The paper is well written, although, it is simple and doesn't really provide much details to complex work done by such committees, I can say from experience that very little is published in this important field of decision making that ultimately affects large number of patients at an institution, so all papers in this field are welcomed to add some enrichment to the field.

Reference 7 and 8 are repeated.

Study design:

Line 110-114 I think this paragraph just repeats what is mentioned in the paragraph proceeding it.

Sample size

Line 117 I am a bit confused with the number of hospitals, under sample size section you mention that invitations were sent to 16 + 47 + 4 institutions, and that only 16 had a MTC, but then in results say that 16 out of 21 responded. Please review this and make it clear to the reader.

Results:

Line 217: it would be good to mention the frequency of the meeting for those with the highest frequency. (Monthly?), table 1 also doesn’t specify the frequency of the meetings.

The number of meetings in table 2 (4 (25) were once every month) is different from the statement in line 251 (56% held meetings bi-monthly)

Reviewer #2: Although the manuscript provides a description of/insights in DTC activities in hospitals who have such a multidisciplinary committee, I'm not convinced that this article offers in its present form the right message for an international public:

1. It is nice to read good intentions in hospitals having DTC structures in place, but the difference between (the characteristics of) hospitals with and without such a team is not well explained. Is there some information on the number of DTCs in non-participating hospitals? To my opinion a real problem might be there.

2. There is always a risk of desirable answers in questionnaires. It is to me a missed opportunity that there were no questions in depth about the change in usage patterns of antibiotics. Were there observed changes due to (better) management or what were the results of these drug use evaluations... To me there is a difference between sending out good guidelines on the one hand and effective changes in usage patterns through management on the other hand. This study does not provide an answer on that.

3. It surprised me that in the context of shortages of antibiotics (worldwide) the questionnaire merely gave answers on more restrictive use rather than on ensuring good supply. To mention, in high income countries there is often a big problem in finding essential antibiotics... and it is then often a question about not depriving patients of treatment of first choice.

In addition some detailed comments:

-Please shorten the abstract

-Is it correct that there are four tertiary hospitals and ten teaching hospitals. Discrepancy needs to be explained for an international audience.

-Abbreviation MTC must be harmonised throughout the manuscript (medicines and therapeutic commitee or medical treatment categorie?)

-Some respondents mention adverse events with certain antibiotics. Were there quality issues with the supplied product?

-Results: characteristics of respondents: not clear what is meant with: "highest frequency" of MTCs? If more regional hospitals answered the questionnaire as opposed to other hospital "types" than it is normal that the number of DTC in your dataset is higher... Numbers need to be weighed against total number of potential hospitals that can respond...

6. PLOS authors have the option to publish the peer review history of their article (what does this mean?). If published, this will include your full peer review and any attached files.

Reviewer #1: **Yes: **Laila Carolina Abu Esba

Reviewer #2: No

---

## [Author Response · Author response to Decision Letter 0]

18 Sep 2023

RESPONSE TO REVIEWER'S COMMENTS ON MANUSCRIPT, PONE-D-23-21276

Dear Editor: 

We appreciate the comments from the editors and reviewers. They have improved the quality of our manuscript. We have revised the submitted manuscript according to the editor and reviewers’ suggestions. We have responded to all comments, questions, and suggestions in detail. We appreciate so much as authors.

Yours, 

Isaac Magulu Kimbowa 

Corresponding author

RESPONSE TO EDITORS AND REVIEWERS COMMENTS PONE-D-23-21276

Editor Comments:

Comments # 01 Editor : 1. Please ensure that your manuscript meets PLOS ONE's style requirements, including those for file naming. The PLOS ONE style templates can be found at 

Response 

We have abided by all PLOS ONE's styles requirements, including filing requirements. 

Comments # 02 Editor : 2. Thank you for stating the following in your Competing Interests section: 

"No author with competing interest"

Response 

We have included in our cover letter that the authors have declared that no competing interest exist 

Comments # 03 Editor: 3. We note that you have stated that you will provide repository information for your data at acceptance. Should your manuscript be accepted for publication, we will hold it until you provide the relevant accession numbers or DOIs necessary to access your data. If you wish to make changes to your Data Availability statement, please describe these changes in your cover letter and we will update your Data Availability statement to reflect the information you provide.

Response 

We have provided accessibility through the link below. The DOI is also provided below 

https://osf.io/4e2cf/?view_only=414c07d6dd4e434eb177272b02cef25f

DOI 10.17605/OSF.IO/4E2CF

Comments # 04 Editor: 4. Please amend your list of authors on the manuscript to ensure that each author is linked to an affiliation. Authors’ affiliations should reflect the institution where the work was done (if authors moved subsequently, you can also list the new affiliation stating “current affiliation:….” as necessary).

Response 

We have corrected the anomaly. 

Comments # 05 Editor: 5. Please include captions for your Supporting Information files at the end of your manuscript, and update any in-text citations to match accordingly. Please see our Supporting Information guidelines for more information: http://journals.plos.org/plosone/s/supporting-information. 

Response 

We have added in-text citation matching Supporting Information guidelines accordingly

Reviewers' comments:

Comments to the Author

1. Is the manuscript technically sound, and do the data support the conclusions?

Reviewer #1: Yes

Reviewer #2: Partly

2. Has the statistical analysis been performed appropriately and rigorously?

Reviewer #1: Yes

Reviewer #2: N/A

3. Have the authors made all data underlying the findings in their manuscript fully available?

Reviewer #1: Yes

Reviewer #2: Yes

4. Is the manuscript presented in an intelligible fashion and written in standard English?

Reviewer #1: Yes

Reviewer #2: No

5. Review Comments to the Author

Reviewer 1

Reviewer 1: Comment # 01: Reviewer #1: Thanks for the authors work on this interesting snapshot of the MTCs structure and function in their country. The paper is well written, although, it is simple and doesn't really provide much details to complex work done by such committees, I can say from experience that very little is published in this important field of decision making that ultimately affects large number of patients at an institution, so all papers in this field are welcomed to add some enrichment to the field.

Response 

Thanks for the comment

Reviewer 1: Comment # 01:Reference 7 and 8 are repeated.

Response 

Thanks for the observation. We have corrected this repetition ( Lines 47, pages 4).

Reviewer 2: Comment # 01:Study design:

Line 110-114 I think this paragraph just repeats what is mentioned in the paragraph proceeding it.

Response 

We have corrected this repetition (Lines 110-114, pages 6). Thanks for the observation

Reviewer 1; Comment # 01:Sample size

Line 117 I am a bit confused with the number of hospitals, under sample size section you mention that invitations were sent to 16 + 47 + 4 institutions, and that only 16 had a MTC, but then in results say that 16 out of 21 responded. Please review this and make it clear to the reader.

Response 

We have improved the clarity (Lines 119 to 121, pages 8). We invited 63 hospitals which included 16 public regional referrals, 47 public general hospitals and four tertiary private not for profit hospitals. Only 21 /67 (31.3%) hospitals reported presence of MTCs. Among the 21 hospitals, only 16 ( ten regional referral, accepted to participate in the study. They granted us administrative clearance. 

Reviewer 1: Comment # 01:Results:

Line 217: it would be good to mention the frequency of the meeting for those with the highest frequency. (Monthly?), table 1 also doesn’t specify the frequency of the meetings.

Response 

Thanks for the suggestion. We have reported in Lines 237, pages 13, MTCs with the highest frequency of meetings. We have reported under MTC administration since its where this variable is most suited. 

Reviewer 1; Comment # 01: The number of meetings in table 2 (4 (25) were once every month) is different from the statement in line 251 (56% held meetings bi-monthly)

Response 

We have improved the clarity of the statement to once every two month and moved the statement to Lines 237, pages 13.

Reviewer 2

Reviewer #2: Although the manuscript provides a description of/insights in DTC activities in hospitals who have such a multidisciplinary committee, I'm not convinced that this article offers in its present form the right message for an international public:

Response 

Several studies have reported on the lack of structures for implementation of antimicrobial stewardship to optimise antibacterial use in hospitals. In Low and middle-income countries, antimicrobial stewardship programmes are not a standalone committee, they exist as subcommittees of MTCs. However, literature on the implementation of MTCs is very limited among LMICs, despite several efforts from World Health Organisation to introduce them. In Uganda, the national action plan for antimicrobial resistance (2018-2023) is under implementation. Its objective four emphasizes the need for optimising antibacterial use through implementing antimicrobial stewardship programmes. It was prudent to investigate whether MTC exist in Uganda and whether their structures have a role in optimising antibacterial use, which our findings have adequately answered. 

Reviewer 2: Comment # 01:It is nice to read good intentions in hospitals having DTC structures in place, but the difference between (the characteristics of) hospitals with and without such a team is not well explained. Is there some information on the number of DTCs in non-participating hospitals? To my opinion a real problem might be there.

Response

It a very nice observation since it would clearly give us an indepth analysis of the differences in optimal antibacterial use between those hospital with or without MTCs. However, we urgently needed an understanding of the role of the MTC structure supporting optimal antibacterial use. We needed to know what structures are depended upon when MTCs are supporting optimal antibacterial use. Its from such a study that we can conduct the study you have in mind. We realise now in Uganda MTCs existed in only 21 out of 67 hospitals. A large proportion of Ugandan hospitals lack this committee which are instrumental on medicines use. 

Reviewer 2: Comment # 02: There is always a risk of desirable answers in questionnaires. It is to me a missed opportunity that there were no questions in depth about the change in usage patterns of antibiotics. Were there observed changes due to (better) management or what were the results of these drug use evaluations... To me there is a difference between sending out good guidelines on the one hand and effective changes in usage patterns through management on the other hand. This study does not provide an answer on that.

Response 

We appreciate you concern. The above concern you raise in your comment we captured the it as unanswered variables in the questionnaire. We have collected data on the issues you have raised and hop to share another manuscript. We shall be glad to review us again whether the manuscript will have answered you correctly

We could not proceed the above concern without responding to whether there are MTCs in Uganda and whether their structure have role in supporting optimal antibacterial use. The findings of the above study can be a foundation of explaining the next study which is assessed by all healthcare provider on the effectiveness of the MTC in improving antibacterial use. 

Reviewer 2: Comment # 03: It surprised me that in the context of shortages of antibiotics (worldwide) the questionnaire merely gave answers on more restrictive use rather than on ensuring good supply. To mention, in high income countries there is often a big problem in finding essential antibiotics... and it is then often a question about not depriving patients of treatment of first choice.

Response 

Ugandan hospitals are still challenged with availability of antibacterials, I agree with the comment.

In the designing of the questionnaire, we were so concerned about the existence of MTC structure. We had to assess whether the structure supported objective selection and evaluation of antibacterials on the formulary. Once you can evaluate and select well the antibacterials, this will support availability and patient access to medicines thus minimise shortages. In our findings this is well reported from MTCs.

We needed to understand what criteria are supporting these MTCs in selecting and evaluating antibacterial for their formulary. Have they increased availability and access to antibacterials in the facility? Indeed our findings have demonstrated the existence of AWaRE categorisation, ABC analysis to track consumption. All these are very instrumental in generating essential antibacterials. Once selected, the MTC reduced stock-out by stocking these few selected antibacterials in large quantities. The question that remained unanswered was whether they were activities to support optimal use of the stocked antibacterials. There was a subcommittee on antimicrobial stewardship that implemented several activities to optimise antibacterial use. They are many follow-up studies that can emerge from this study on improving optimal antibacterial use in hospitals.

In addition some detailed comments:

Reviewer 2: Comment # 04: -Please shorten the abstract

Response 

We have shortened the abstract 

Reviewer 2: Comment # 05: -Is it correct that there are four tertiary hospitals and ten teaching hospitals. Discrepancy needs to be explained for an international audience.

Response 

We had 13 tertiary hospitals comprising of 10 regional referral and three PNFP that participated in the study. The same tertiary hospitals are referred to as teaching hospitals. 

Reviewer 2: Comment # 06: -Abbreviation MTC must be harmonised throughout the manuscript (medicines and therapeutic commitee or medical treatment categorie?)

Response 

We have corrected this abbreviation on Lines 334, pages 20 

6. PLOS authors have the option to publish the peer review history of their article (what does this mean?). If published, this will include your full peer review and any attached files.

Do you want your identity to be public for this peer review? For information about this choice, including consent withdrawal, please see our Privacy Policy.

Reviewer #1: Yes: Laila Carolina Abu Esba

Reviewer #2: No

---

## [Editor Report · Decision Letter 1]

22 Nov 2023

The role of medicines and therapeutics committees structure in supporting optimal antibacterial use in hospitals in Uganda: A mixed method study

PONE-D-23-21276R1

Dear Dr. Kimbowa,

We’re pleased to inform you that your manuscript has been judged scientifically suitable for publication and will be formally accepted for publication once it meets all outstanding technical requirements.

Kind regards,

Joseph Olusesan Fadare

Academic Editor

PLOS ONE

Additional Editor Comments (optional):

Thank you for addressing to comments raised.
---

## [Editor Report · Acceptance letter]

6 Dec 2023

PONE-D-23-21276R1 

The role of medicines and therapeutics committees structure in supporting optimal antibacterial use in hospitals in Uganda: A mixed method study 

Dear Dr. Kimbowa:

I'm pleased to inform you that your manuscript has been deemed suitable for publication in PLOS ONE. Congratulations! Your manuscript is now with our production department. 

Kind regards, 

on behalf of

Dr. Joseph Olusesan Fadare 

Academic Editor

PLOS ONE